# Homogeneous surrogate virus neutralization assay to rapidly assess neutralization activity of anti-SARS-CoV-2 antibodies

Sun Jin Kim[1,20], Zhong Yao[2,20], Morgan C. Marsh[1], Debra M. Eckert[3], Michael S. Kay[3], Anna Lyakisheva[2], Maria Pasic[4,5,6], Aiyush Bansal[6], Chaim Birnboim[6], Prabhat Jha[6], Yannick Galipeau[7], Marc-André Langlois [7,8], Julio C. Delgado[9,10], Marc G. Elgort[9], Robert A. Campbell[10,11,12], Elizabeth A. Middleton[11,12,13], Igor Stagljar [2,14,15,16,17✉] & Shawn C. Owen [1,18,19✉]

The COVID-19 pandemic triggered the development of numerous diagnostic tools to monitor infection and to determine immune response. Although assays to measure binding antibodies against SARS-CoV-2 are widely available, more specific tests measuring neutralization activities of antibodies are immediately needed to quantify the extent and duration of protection that results from infection or vaccination. We previously developed a 'Serological Assay based on a Tri-part split-NanoLuc® (SATiN)' to detect antibodies that bind to the spike (S) protein of SARS-CoV-2. Here, we expand on our previous work and describe a reconfigured version of the SATiN assay, called Neutralization SATiN (Neu-SATiN), which measures neutralization activity of antibodies directly from convalescent or vaccinated sera. The results obtained with our assay and other neutralization assays are comparable but with significantly shorter preparation and run time for Neu-SATiN. As the assay is modular, we further demonstrate that Neu-SATiN enables rapid assessment of the effectiveness of vaccines and level of protection against existing SARS-CoV-2 variants of concern and can therefore be readily adapted for emerging variants.

[1] Department of Pharmaceutics and Pharmaceutical Chemistry, University of Utah, Salt Lake City, UT, USA. [2] Donnelly Centre, University of Toronto, Ontario, Canada. [3] Department of Biochemistry, University of Utah School of Medicine, Salt Lake City, UT, USA. [4] Department of Laboratory Medicine & Pathobiology, University of Toronto, Toronto, Ontario, Canada. [5] Department of Laboratory Medicine, St. Joseph's Health Centre, Toronto, Ontario, Canada. [6] Centre for Global Health Research, Unity Health Toronto, University of Toronto, Toronto, Canada. [7] Department of Biochemistry, Microbiology & Immunology, Faculty of Medicine, University of Ottawa, Ottawa, Canada. [8] University of Ottawa Centre for Infection, Immunity and Inflammation (CI3), Ottawa, Canada. [9] ARUP Institute for Clinical and Experimental Pathology, Salt Lake City, UT, USA. [10] Department of Pathology, University of Utah, Salt Lake City, UT, USA. [11] Department of Internal Medicine, Division of General Medicine, University of Utah, Salt Lake City, UT, USA. [12] Molecular Medicine Program, University of Utah, Salt Lake City, UT, USA. [13] Department of Internal Medicine, Division of Pulmonary and Critical Care Medicine, University of Utah, Salt Lake City, UT, USA. [14] Department of Biochemistry, University of Toronto, Ontario, Canada. [15] Department of Molecular Genetics, University of Toronto, Ontario, Canada. [16] Mediterranean Institute for Life Sciences, Meštrovićevo Šetalište 45, Split, Croatia. [17] School of Medicine, University of Split, Split, Croatia. [18] Department of Biomedical Engineering, University of Utah, Salt Lake City, UT, USA. [19] Department of Medicinal Chemistry, University of Utah, Salt Lake City, UT, USA. [20] These authors contributed equally: Sun Jin Kim, Zhong Yao. ✉email: igor.stagljar@utoronto.ca; shawn.owen@hsc.utah.edu

SARS-CoV-2 continues to threaten the world's health as emerging variants of concern have the potential to circumvent deployed vaccines. Simple and rapid SARS-CoV-2 serological tests are needed to accurately measure the level and duration of neutralization activity of antibodies that arise from natural infection or vaccination. Currently, there are several FDA-approved serological tests under Emergency Use Authorizations (EUA), many of which can detect IgM or IgG against SARS-CoV-2, but do not measure their neutralization efficacy specifically[1–3]. Functional neutralizing antibody titers are often measured with pseudotyped viruses, however, long assay time and discrepancies in published assay protocols have limited their use[4–6]. Alternatively, surrogate virus neutralization assays have been developed to circumvent the use of pseudovirions[7–10]. Although some of these assays have shown successful measurement for serosurveillance of clinical samples, they often resemble ELISA, requiring multiple time-consuming binding and washing steps, while others have not yet reported successful measurement of clinical samples, likely due to the instability of recombinant proteins in serum and plasma. Here, we report the development of a homogeneous surrogate virus neutralization assay (hsVNA) called "Neu-SATiN" by reconfiguring our previously designed serological assay (Serological Assay based on split Tri-part Nanoluciferase; SATiN)[11] to quantify the degree of neutralization from antibodies directly from plasma or serum. SATiN is a serological assay based on a protein-protein interaction approach for detection of IgGs against the Spike protein of SARS-CoV-2[12].

In our previous report, we developed SATiN as a homogeneous serological assay platform that utilizes a tri-part NanoLuc®, which is split into two small peptide tags, β9 and β10 (each about 1 kDa), and one large fragment, Δ11S (18 kDa). The SATiN assay utilizes spike protein and Protein G tagged with either β9 or β10. Upon simultaneous binding of the tagged spike protein and Protein G to anti-SARS-CoV-2 antibody, β9 and β10 are brought into proximity which induces refolding of Δ11S into active luciferase, producing glow-type luminescence[11]. Using the same tri-part NanoLuc®, we now show the development of a homogeneous neutralization assay version of SATiN (Neu-SATiN). In Neu-SATiN, enzyme fragment peptides, β9 or β10, are fused to the ACE2 receptor, the target of infection, and to the SARS-CoV-2 spike (S) protein. We hypothesized that when the ACE2 and the S proteins interact, the fused split-NanoLuc® fragments are driven to within ~100 Å of each other, allowing Δ11S to reconstitute into fully functional NanoLuc®. Importantly, when the interaction between the ACE2 and S proteins is blocked by neutralizing antibodies, the tags are prevented from interacting and

subsequent complementation of NanoLuc® is blocked (Fig. 1a). Therefore, in Neu-SATiN, the level of neutralization correlates with the decrease of luminescence. As the assay is intentionally modular, full-length ectoderm of the S protein of SARS-CoV-2 variants of concern can be quickly produced and swapped with wild-type S protein to assess immunological protection. Moreover, as Neu-SATiN is designed to be a "mix-and-read" assay that is performed at the conventional lab bench, the actual hands-on time is <30 min, significantly improving turnaround time.

## Results

**Protein fusion design and testing.** We considered three options for fusing the split-luciferase fragments to the SARS-CoV-2 spike (S) protein and its target, the human ACE2 receptor: (1) N-terminus, (2) C-terminus, or (3) both termini. Although surface loops are possible fusion points, modifying these domains may interfere with neutralizing antibody binding. Based on our previous experience, we know that reconstitution of split-NanoLuc® is most efficient when the fusion locations are within ~50–100 Å[13–15]. We used molecular modeling to determine possible fusion points on S protein and ACE2 that place fragments within this proximity (Figs. 2a and 3a).

To test feasibility, we initially built the system using only the receptor-binding domain (RBD) of the SARS-CoV-2 spike protein and ACE2. Based on molecular modeling of RBD and ACE2 (Fig. 2a)[16], the distance from the N-terminus of ACE2 to the N-terminus of RBD is ~60 Å and to the C-terminus is ~53 Å (PDB ID: 6M0J). In contrast, the C-terminus of ACE2 is more than 100 Å away from either terminus of RBD and thus was excluded as a potential tag fusion site. The complete list of binders used in this study is listed in Supplementary Table 2. Using purified recombinant proteins, we confirmed that the engineered RBD and ACE2 binders with complementary tags produced detectable luminescent signal when combined with the large fragment of the split luciferase, Δ11S, in human serum. Each binder combination shows substantial luminescence; however, there are differences in signal-to-background ratios (Supplementary Fig. 1a). We also confirmed substantial decrease in the signal when the binders were incubated with neutralizing antibody (NAb) (Sino Biological, 40592-MM57), indicating inhibition of RBD binding to ACE2 to prevent complementation of the split-luciferase fragments. For any given pair, the fractional decrease in luminescent signal as a function of increasing concentration of NAb displays a typical dose–response curve for inhibition that is specific to the NAb (Supplementary Fig. 1b). This suggests that

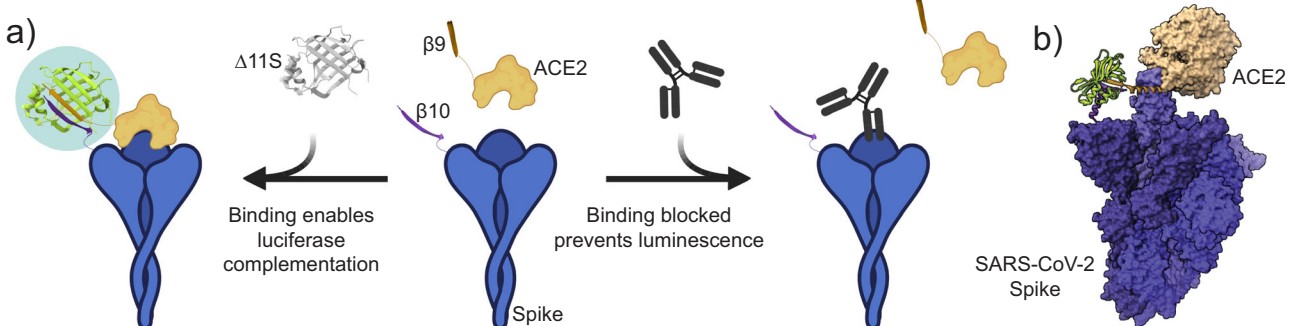

**Fig. 1 General schematic of the Neu-SATiN COVID-19 neutralization assay and molecular modeling of spike (S) protein and ACE2 interaction. a** Tri-part NanoLuc® peptide fragments are individually fused to recombinant S protein (purple) and ACE2 (tan). Interaction of S protein and ACE2 induces complementation of the split-luciferase and 'turns on' luminescence (left). In the presence of neutralizing antibodies, the interaction between S protein and ACE2 is blocked, preventing luminescence (right). Figure generated using BioRender. **b** Molecular model of the predicted refolding of NanoLuc® (PDB ID: 5IBO) is shown (green) after complementation of fragments β10 and β9 is driven by the interaction between full spike protein (trimer) and ACE2 (PDB ID: 7A97).

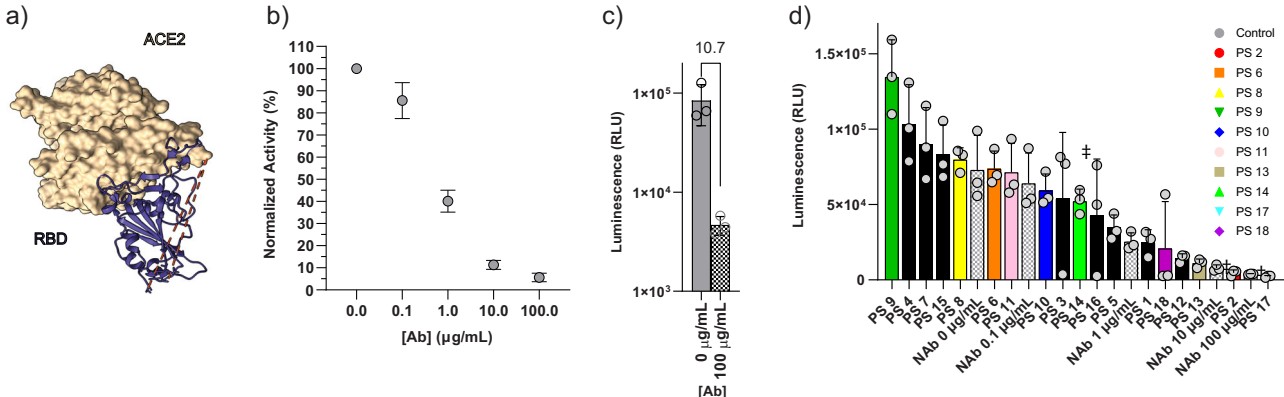

**Fig. 2 Binder pair screening in human serum and patient samples (PS 1–18). a** Molecular modeling of the distances between N-terminus of ACE2 to the N-terminus of RBD is ~60 Å and to the C-terminus is ~53 Å (PDB ID: 6M0J). As such, luciferase fragments can be fused at either terminus of RBD and N-terminus of ACE2. **b** (S)RBD-β9 and β10-ACE2 binder pair was screened with increasing concentrations of neutralizing Ab (NAb, Sino Biological 40592-MM57) in human serum. **c** Using (S)RBD-β9 and β10-ACE2 binder pair, the signal between 0 μg/mL of NAb (darker color) vs. 100 μg/mL of NAb (lighter color adjacent bar) shows a 10-fold decrease upon neutralization. **d** Testing 18 patient plasma samples: (S)RBD-β9 and β10-ACE2 were mixed directly with plasma samples, followed by the addition of the detection solution (Δ11S and substrate). Samples 16, 17, and 18 (indicated by ‡ above bar) are known to be convalescent whereas other samples are from ICU patients with unknown antibody presence/levels. Source data are provided as a Source data file.

although the maximum signal intensity produced by each binder pair is different, the proportional decrease in the signal in the presence of NAb is similar. Therefore, the ability to measure NAb antiviral activity is feasible with several different pairs. We characterized binder pairs further by calculating $IC_{50}$ of the NAb using each pair and stability of the complex in high concentrations of non-binding IgG (Supplementary Fig. 1c, d). We selected the pair (S)RBD-β9 and β10-ACE2 based on the lowest discrepancy in $IC_{50}$ value compared to the manufacturer's report and in resolving concentration to response.

**Evaluation of Neu-SATiN on convalescent samples**. After validating Neu-SATiN's ability to detect NAb activity, we validated the assay with clinical plasma samples from actively infected (from ICU) and convalescent patients ($n = 18$). The plasma samples were tested and binned into 5 different groups in respect to their relative luminescence compared to the control NAb (Fig. 2d). Of the convalescent patient samples (PS 16, 17, 18), patient sample 17 (PS 17) showed inhibition similar to 100 μg/mL of the control NAb. PS 16 and PS 18 were not as effective as PS 17, but still showed significant decrease in luminescence intensity compared to 0 μg/mL of NAb. As the neutralization activity depends on the epitope, affinity and concentration of the antibodies, the comparison between the control antibody and the antibodies in patient plasma is only relative. Nevertheless, our data confirm that Neu-SATiN is able to distinguish the presence or absence of antiviral antibodies and quantify the relative level of neutralization directly in clinical samples.

We compared Neu-SATiN to a pseudovirus neutralization (PSV) assay based on a human immunodeficiency virus (HIV) system with luciferase gene reporter (Supplementary Fig. 1e, f). The results from Neu-SATiN and PSV assay show good correlation at highest antibody concentrations (Extended Fig. 1e, f) with a Pearson's $r$ value of 0.86. Correlation is slightly lower across all IgG concentrations (Supplementary Fig. 1e) with a Pearson's $r$ value of 0.81. Both PSV assay and Neu-SATiN showed that PS 17 had the highest inhibition activity, and this was similar to the positive control (human serum spiked with 100 μg/mL NAb).

**Neu-SATiN full-length S protein evaluation of serum samples with differing levels of virus protection**. Even though many of

the antibodies produced by vaccines target the RBD of S protein[17,18], the more pressing concern is protection against variants of concern which often have mutations outside of the RBD[19,20]. In fact, all currently approved vaccines in US and Canada are based on full-length S protein sequences and have been generated before the onset of variants in 2021. As such, we expanded Neu-SATiN using ACE2 and the full ectodomain of S protein, trimerized through the foldon domain, including known amino acid substitutions to stabilize conformation[21]. We hypothesized that using the full, trimerized S protein should provide a more comprehensive measurement of the neutralization effect. Molecular modeling was used again to examine the distances between the termini of two proteins (Fig. 3a) using PDB ID: 7A97[22]. Based on this analysis, we decided to fuse β10 to the N-terminus of S protein and β9 to the N-terminus of ACE2. The length of linker connecting the proteins to the tags was also doubled from 9 to 18 amino acids to increase the flexibility of the tags and to encompass a broader range of potential binding confirmations. Consistent with previous studies[23], we found the interaction between S ectodomain and ACE2 monomer is relatively low, likely due to ACE2 instability, and decided to use a dimerized ACE2 binder created by fusing the ACE2 N-terminal domain (a.a. 16–614) with the human IgG Fc fragment (Supplementary Table 1). These two binders, denoted as β10-(S)-WT and β9-ACE2-Fc, were validated for binding-induced luminescence (Fig. 3b). The full wild-type spike protein (WT) version of the binder pair shows average luminescence signal of $1.6 \times 10^6$ RLU indicating successful complementation of the split-Nano-Luc® fragments and shows a robust 3000-fold signal-to-background ratio. We produced mutated S proteins for each of the major SARS-CoV-2 variants of concern: Alpha, Beta, Gamma, Delta, and Omicron[19,20] (mutation sequences in Supplementary Table 2). The mutations were made throughout the ectodomain of S protein and not just limited to the RBD. Additional mutations were made to the furin cleavage site to prevent unwanted degradation of the binders by cell culture proteases and to enable purification of recombinant proteins[24,25]. The orientation of the tags on each protein was kept the same: β10-(S)Variant and β9-ACE2-Fc. All five variants (Alpha, Beta, Gamma, Delta, and Omicron) produced distinct luminescence signals with high signal-to-background (Fig. 3b). The neutralization of the WT pair (WT S protein with ACE2) corresponds well with concentrations

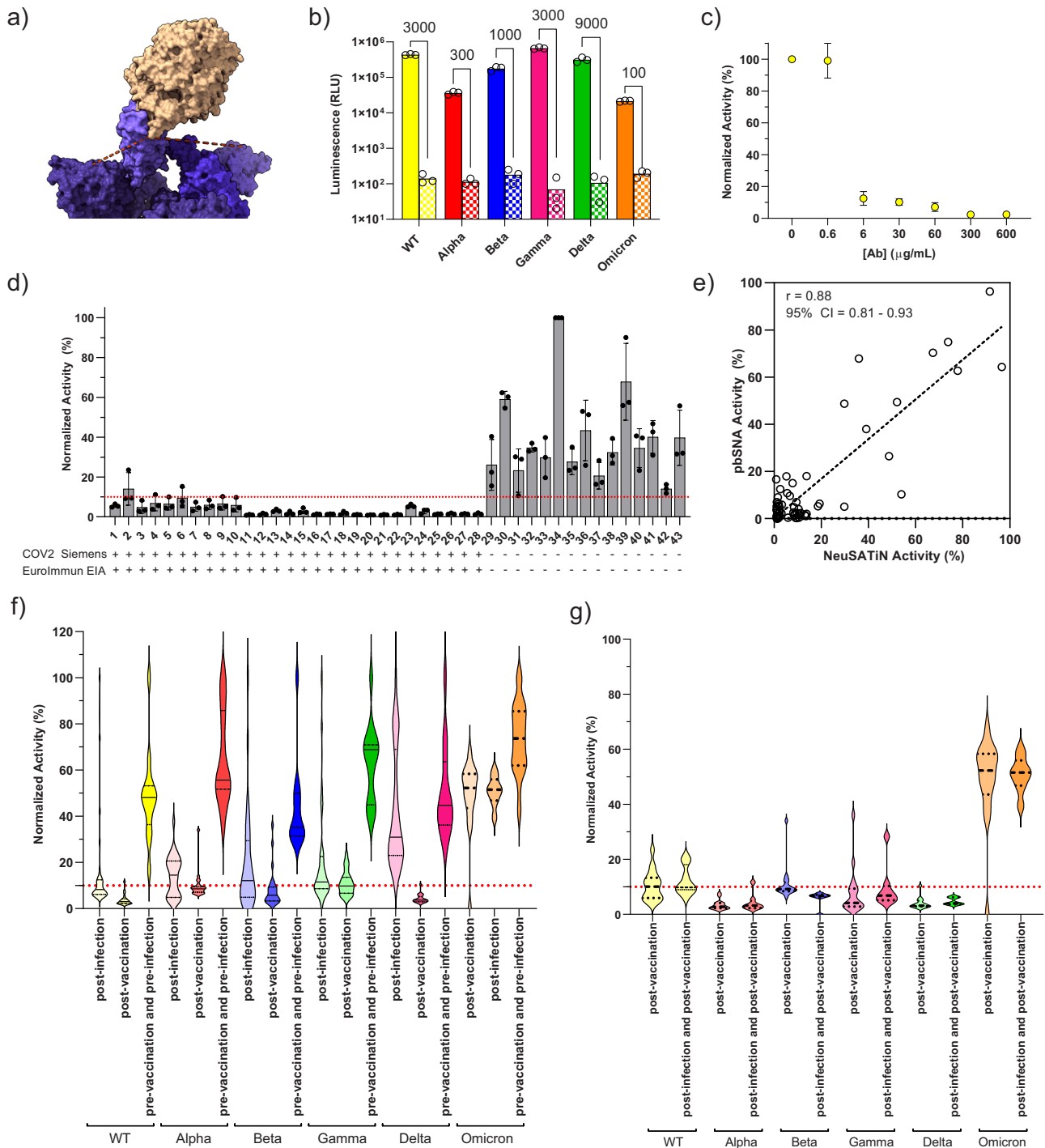

of a NAb (Sino Biological, 40592-R001) (Fig. 3c); however, there was limited neutralization of most of the variants using this NAb (Supplementary Fig. 2).

We obtained another set of serum samples ($n = 43$) that have been tested with two different Emergency Use Authorized COVID-19 S protein-binding antibody assays: (1) COV2G Siemens 1st Gen (target antigen: RBD, cutoff: ≥1.0), and (2) EUROIMMUN (target antigen: S1, cutoff: ≥1.1). The samples can be further categorized into three different groups: (1) SARS-CoV-2 exposed with no vaccination history (post-infection, Samples 1–18), (2) SARS-CoV-2 exposed with vaccination history (post-infection and post-vaccination, Samples 19–28), and (3) neither exposed nor vaccinated (no infection and no vaccination, Samples

29–43). All of the samples were screened with our full-length-WT S protein Neu-SATiN for the presence of neutralizing antibodies. Commercial human serum spiked with NAb (Sino Biological, 40592-R001) was used as a positive control and to determine relative neutralization activity of the patient samples.

Samples with previous infection history (Samples 1–28, $n = 28$) showed normalized activity score <10% (Fig. 3d—indicated by red dotted line) in Neu-SATiN, with the exception of Sample 2 (frequency = 27/28, 96%). These were also the samples that scored >20.00 in Siemens 1st Gen and >8 in EUROIMMUN EIA, both indicating the presence of anti-SARS-CoV-2 antibodies against the WT S protein (Supplementary Table 3). Intriguingly, when the patients were previously infected and then vaccinated

**Fig. 3 Validation of full spike proteins (wild type and variants) and ACE2 binders in the serosurveillance of clinical samples. a** Molecular modeling of the distances between N-terminus of ACE2 to the N-termini of nearest S protein is ~48 Å and ~88 Å, respectively (PDB ID: 7A97). **b** Comparison of signal and background for the full spike (wild type and variants) and ACE2 pairs. **c** Serial dilution of a commercially available NAb (Sino Biological, 40592-R001) in the presence of β10-(S)WT and β9-ACE2-Fc pair. **d** Serum samples that have been tested previously on two different COVID-19 detection assays were also tested using Neu-SATiN. Wild type full spike protein with β10 tag (β10-(S)WT) and ACE2 with β9 tag (β9-ACE2-Fc) was used as the binders. Signals from patient samples were normalized to the signal from normal human serum with the binders alone (no neutralizing antibodies). The red dotted line at 10% activity (i.e., 90% neutralization) indicates the cutoff line to distinguish neutralizing samples from the non-neutralizing samples; '+' and '−' signs below the sample number indicate the result from two prior tests detecting anti-SARS-CoV-2 antibodies (COV2G Siemens 1st Gen and EUROIMMUN EIA). **e** Activity measured in protein-based surrogate neutralization assay (pbSNA) versus neu-SATiN for the wild-type (WT) variant shows high correlation with a Pearson's correlation test $r$ value of 0.88. $N = 66$. **f** Neutralization efficacy of patient samples against WT and variant S proteins. Samples used in (**d**) ($n = 43$) were tested with an additional $n = 35$ patient samples that were positive for anti-SARS-CoV-2 antibodies. Luminescence signals from each patient serum were measured and normalized to human serum (no NAb). Known post-vaccination, post-infection, and negative (no known infection or vaccination) samples were plotted separately. Red dotted line at 10% indicates the cutoff between the negative samples and the positive samples for distinction of neutralization. **g** Comparison of serum data from patients that were vaccinated to patients that were vaccinated after a documented infection ($n = 40$). All patient samples were collected before November 2021, prior to any known Omicron infections. Source data are provided as a Source data file.

(Samples 19–28, $n = 10$), their sera showed normalized activity scores of 5% or below (frequency = 10/10, 100%). The various levels of neutralization shown in Samples 1–18 are possibly due to difference in sample collection day post-infection, as it is known that antibody levels are highest 4–5 weeks after symptom onset[26,27]. Conversely, all SARS-CoV-2 negative patients (Samples 29–43, $n = 15$), based on standard antibody test, showed signals above 10% (frequency = 15/15, 100%), indicating no neutralization activity. Further, the activity of Neu-SATiN was benchmarked against a protein-based neutralization assay (Supplementary Fig. 3) of the wild-type variant spike protein, showing good correlation with a Pearson's $r$ value of 0.88 ($N = 66$) (Fig. 3e) and of the Omicron variant spike protein (Supplementary Fig. 4). The protein-based neutralization assay is performed by coating microwells with recombinant SARS-CoV-2 Spike trimer and then adding plasma from patient samples. An ACE2-HRP construct is used to assess the presence of neutralizing antibodies that prevent binding of ACE2 to Spike trimer.

Altogether, the data supports that our newly developed Neu-SATiN neutralization assay is a reliable surrogate test that shows congruent results compared to already established tests. As we recognize that immunities generated by vaccination versus infection are known to produce antibodies targeting different parts of S protein[28] and that the protection against variants based on vaccination and/or previous infection is variable, we tested the same patient samples used to validate β10-(S)WT/ β9-ACE2-Fc, along with an additional set of infected patients ($n = 35$), for the samples' neutralization efficacies towards different variants of concern. Since a universal NAb that can neutralize WT and all variants is not available, we were not able to determine relative degree of neutralization compared to a known concentration of NAb for each of the variants, but only for the WT. Instead, we compared fractional decrease in luminescence signal from the positive samples to negative samples for each variant pair; the highest signal observed from the negative group was considered 100% activity (i.e., no decrease in signal) and subsequent decrease in signal was determined for the positive groups. As shown in Fig. 3f, the NAb negative samples (darkest color violin plots) tend to display a wide range of signal compared to the NAb positive groups. Overall, the mean fractional signal observed from the positive samples (post-infection or post-vaccination) tested either with WT or variant pairs were 10% or lower. In other words, the signal measured from the positive samples were <10% of the signal from the negative group. This suggests that most of the samples within the positive groups have some level of neutralization ability towards each of the variant SARS-CoV-2s. In particular, both WT and Gamma variant were neutralized

almost fully by immunity generated by vaccination; however, there appears to be a subset of samples within the Delta, Alpha, and Beta variant groups with minimal neutralization even after infection and/or vaccination (Fig. 3g). It was also notable that natural immunity generated by infection with SARS-CoV-2 was not sufficient to neutralize the Gamma variant and that vaccination was required for stronger neutralization and that almost all samples were ineffective at neutralizing the Omicron variants. These observations are potentially due to antibodies produced by immunities targeting different parts of S proteins can cause differences in recognizing mutated sites[28,29]. Another potential explanation is that the infective strains are also unknown and could induce different responses. Without further information on which vaccines were received by these patients or the sample collection dates post onset of symptoms, we are not able to determine the exact correlation between vaccination and difference in protection against variants.

**Potency of NAb determination by Neu-SATiN.** To expand on our observations, we tested serial dilutions of FDA EUA-approved neutralizing antibodies Regn10933 (casirivimab), Regn10987 (imdevimab), and JS016 (etesevimab), as well as sera with known vaccination history ($n = 24$). Having such data is informative in detecting the lowest effective concentrations (titers) and therefore determining the potency of NAbs. Of the FDA EUA antibodies, imdevimab was the most effective against WT, Alpha, Beta, Gamma, and Delta variants of the S protein. The other two antibodies, casirivimab and etesevimab, were most potent against WT but showed variable potency against variant S proteins. There was no effective neutralization observed for the Beta strain with either antibody (Fig. 4a) and none of these antibodies display any potency to inhibit the Omicron variant, suggesting their limitation in fighting the current pandemic of Omicron dominancy. For the serosurveillance of patient samples (Fig. 4b), no neutralizing activity was observed in patients that were not vaccinated, as expected. The patients with one dose showed negligible potency compared to the patients with two doses with samples collected <50 days after the last dose. For samples collected >50 days after receiving two doses, the potency of antibodies significantly reduced as indicated by shift in titration curve towards the right, consistent with recent reports of waning immunity for some vaccines[30]. In quantifying these patient samples in terms of the 50% neutralization titer (Fig. 4c), the majority show strong neutralization against WT and the least potency against Beta variant, with the other variants of concern intermediate.

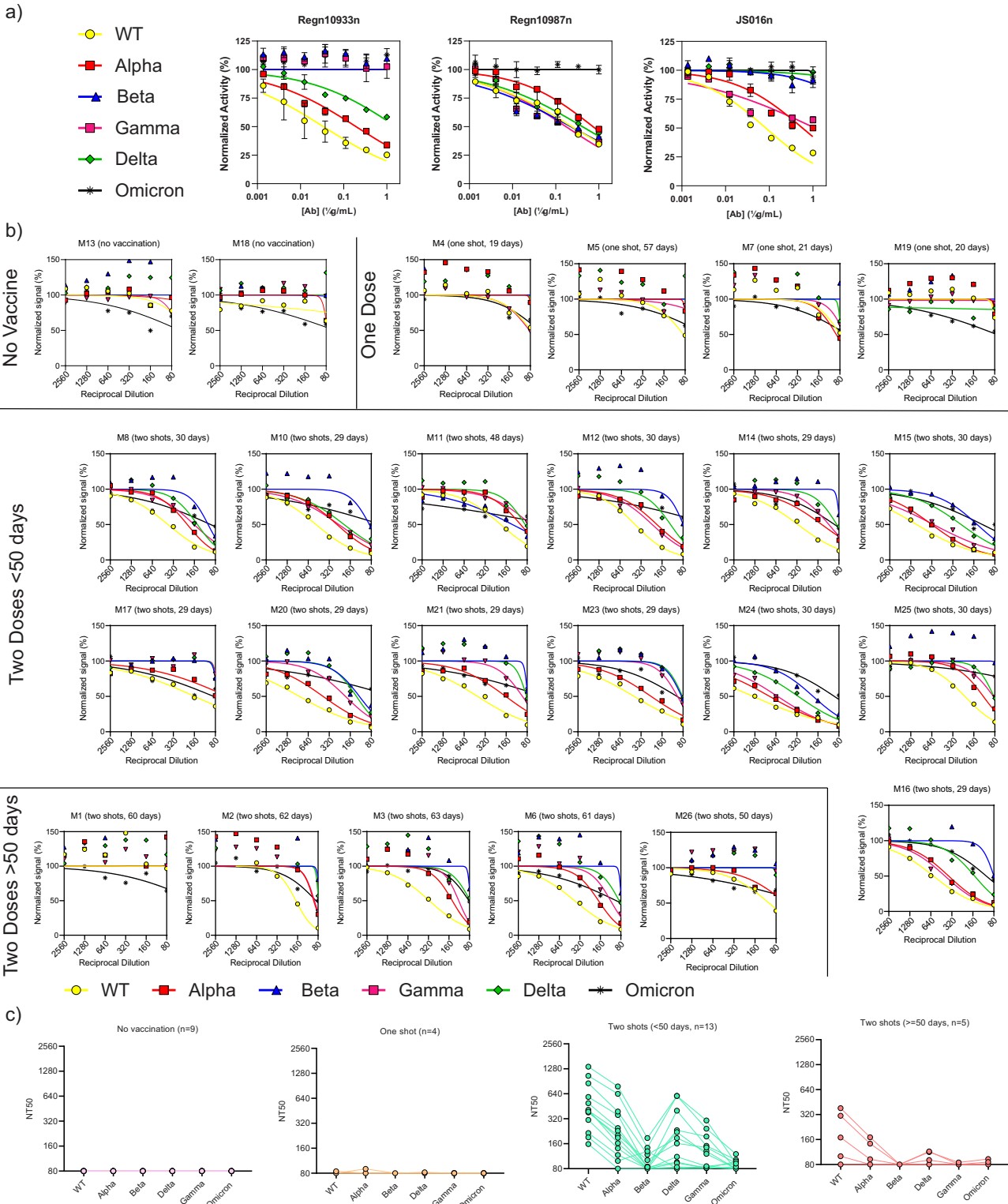

**Fig. 4 Assessing the potency of neutralizing antibodies. a** Neutralizing capabilities of FDA EUA-approved therapeutic antibodies Regn10933 (left, casirivimab), Regn10987 (center, imdevimab), and JS016 (right, etesevimab) were evaluated using Neu-SATiN against different S variants. **b** Evaluation of neutralizing antibodies in individuals with unvaccinated sera ($n = 2$), with one vaccination dose ($n = 4$), with two vaccination doses and collected within 50 days after the second shot ($n = 13$, intervals between shots were 21–37 days), and with two vaccination doses collected more than 50 days after the second shot ($n = 5$, intervals between shots were 21–36 days). **c** Aggregated data showing titers at 50% neutralization (NT50) for each group. Source data are provided as a Source data file.

## Discussion

Since the outbreak of the COVID-19 pandemic, the importance of a virus neutralization assay has risen. Virus neutralization assays are the main tools for developing vaccine and therapeutic strategies[31–33]. Although the PSV assay is effective in measuring the degree of infection, maintaining cell cultures and making pseudovirus particles are labor intensive with potential safety concerns[34]. Moreover, batch-to-batch variability in virus production and cell transfection efficiency limit standardization and robust assay results[35]. Numerous immunoassays for rapid detection of anti-SARS-CoV-2 antibodies have been developed, however, these assays mainly focus on capturing and detecting antibodies binding to virus proteins (e.g., spike or nucleocapsid)[2]. Since it is well known that mere binding does not necessarily imply neutralization[36], a true neutralization assay for the better understanding of protective immunity against SARS-CoV-2 is needed. To circumvent the use of virus particles and cells, surrogate versions of neutralization assays have been developed[7–9]. These assays often use ELISA or similar platforms, with multiple time-consuming binding and washing steps, preventing high-throughput screening[37]. In need of an accurate yet more rapid virus neutralization assay, we developed Neu-SATiN as a homogeneous surrogate virus neutralization assay (hsVNA), using a split-luciferase system.

There have been several reports describing efforts to develop surrogate virus neutralization (sVNA) assays for the detection of antibodies against SARS-CoV-2[38–41]. These reports show capabilities as surrogate assays; however, many of these platforms require chemical modification and conjugation of proteins to beads or other solid surfaces, requiring intermediate purification and/or washing steps to remove unbound reagents and analytes. In contrast, Huang et al. reported the development of an hsVNA using the binary version of split-NanoLuc®. Although the approach is similar to the platform we report here, the authors used a monomeric or dimeric form of S1 domain of spike protein tagged with β10 (SmBiT), paired with ACE2 (LgBiT, NanoLuc® beta strands 1 through 9)[38], which was shown to have limited detection of non-RBD targeting antibodies. The importance of conformational changes in spike protein-altering ACE2 binding has been emphasized in many studies[22,42]. Also, with new variants of concern emerging, many of the deployed vaccines and second-wave vaccines in development are targeting full spike protein[43–45]. We initially designed our assay by investigating the interactions between ACE2 and RBD[10]. The result is in agreement with reports that neutralization activities of NAb mostly come from anti-RBD immunoglobulins[36,46]; however, immunologic response can also produce antibodies against epitopes outside of the RBD and may require the trimeric structure of the full spike protein for binding. This compelled us to develop stabilized, trimeric, full-length spike (both WT and major variants of concern) versions of the binders and stable, dimerized ACE2 receptor. The data presented in Fig. 3 demonstrates that β10-(S)WT/β9-ACE2-Fc can differentiate the degree of neutralization directly from patient sera and that variant versions of the spike binders can distinguish anti-SARS-CoV-2 NAb positive serum samples from negative samples (Fig. 3e, f). In addition, Neu-SATiN can provide quantitative analysis of NT50 and thus enables the measurement of potency of anti-SARS-CoV-2 antibodies against different strains (Fig. 4). Combined, this provides a comprehensive screen of a patient's level of protection against the current variants of concern. As the assay is modular, emerging variants of interest can be quickly produced and incorporated as we have demonstrated with the inclusion of the Omicron variant that become prominent during the initial review of this manuscript. As all the patient samples were collected before November 2021, prior to any known wide-spread Omicron infections, our results also suggest that prior infection/vaccination does not provide significant neutralizing protection against the variant.

The results obtained with Neu-SATiN correlate with PSV assays and other antigen-based assays in detecting the neutralization potential of antibodies in clinical samples. It is important to note that natural immunity can produce antibodies that bind several antigens and function through alternate mechanisms, including antibody-dependent phagocytosis (ADP) or antibody-dependent cellular cytotoxicity (ADCC). Although the current format of Neu-SATiN cannot measure these types of antiviral activity, the majority of vaccines use Spike domains[17,18]; therefore, this assay can be used to assess protection developed from immunization. One advantage of Neu-SATiN is that it can be performed homogenously and directly using plasma or serum, which significantly reduces hands-on assay time to <30 min and can be run under standard lab conditions. Importantly, we have demonstrated that the split-NanoLuc® based Neu-SATiN can be applied to full-length spike proteins of the original strain and variants to test neutralization levels of convalescent patient sera. Having a modular technology as a surrogate assay that can be easily adopted as a point-of-care tool is important in tracing and adapting to the evolution of the current pandemic.

## Methods

**Ethical statement**. Studies were approved by relevant ethical boards: ARUP/University of Utah Institutional Review Board (IRB-approved protocol 0007740) and Office of Environmental Health and Safety at the University of Toronto (REB-approved protocol REB 20-107). Informed consent was obtained for participants and no compensation was provided.

**Cell lines, cell culture media and antibodies, and cloning reagents**. The HEK293 cell line was provided by Prof. Jason Moffat at University of Toronto and was originally purchased from ATCC (CRL-1573). CaLu-3 cells were purchased from ATCC (HTB-55). HEK293 cell culture reagents were purchased from Thermo Fisher Scientific. Cloning reagents were purchased from NEB. Two neutralizing antibodies: 40592-MM57 (used for RBD pair screening) and 40592-R001 (used for WT pair screening) were purchased from Sino Biological. Regn10933 (CPC511A), Regn10987 (CPC512A), and JS016 (CPC516A) were purchased from Cell Sciences. Various antibody dilutions were used in this work, with final concentrations in assays ranging from 0 to 600 µg/mL. Specific concentrations can be found within the Figures and corresponding Figure legends.

**Clinical samples**. Samples were obtained from either the University of Utah School of Medicine, ARUP Laboratories, or from Unity Health. University of Utah School of Medicine (total $n = 16$) were obtained from infected patients within 48 h of admission to ICU ($n = 13$) or within 3–5 weeks of positive PCR test for convalescent patients ($n = 3$). Samples for determining neutralization (total $n = 63$) in uninfected, infected, and/or vaccinated were generously provided by ARUP Laboratories (IRB-approved protocol 0007740). Samples for antibody titer and serosurveillance studies (total $n = 24$) were collected from Unity Health employees, enrolled through REB-approved protocol REB 20-107, Toronto. All samples were deidentified.

**Vector construction and transient transfection in HEK293 cells**. Vector cloning for the binder expression was performed as previously reported[11]. All cDNA were cloned into an in-house mammalian expression vector derived from pCMV5 with a signal peptide sequence appended at the N-termini and an octa-histidine stretch at the C-termini. For the constructs that had the tag at the N-termini, either β9 or β10 sequence was placed after the signal peptide, followed by the binder sequence. Likewise, the constructs with C-terminus tag had β9 or β10 sequence right before octa-histidine (Supplementary Table 1). The final products were transfected into HEK293 cells using polyethylenimine Max (Polysciences).

**Production and purification of binders**. HEK293 cells transfected with binder constructs were cultured in DMEM supplied with 10% FBS and 1X antimycotic-antibiotic mixture. Typically, cells were seeded at 50% confluency, and the media was collected every day until the cells became fully confluent. The collected media was filtered through 0.22 µm PES filter before purification. Purification was done on AKTA FPLC using HisPur™ Cobalt Resin. Tween-20 (0.01%), trehalose (0.1%), and glycerol (10%) were added to the final product and kept at −80 °C before use.

**Reconstitution of split-NanoLuc® driven by (S)RBD and ACE2 interaction**. For the screening of (S)RBD binders with ACE2 binders, 10 µL of commercial human serum (Sigma-Aldrich®, S1-100ML) spiked with titrating concentrations from 0 µg/mL to 100 µg/mL of the neutralizing antibody (Sino Biological, 40592-MM57) was combined with 5 µL of (S)RBD binders (10 pmol) and 5 µL of ACE2 binders (10 pmol), and incubated for 30 min. The incubation was done using white, round bottom 96-well plates at room temperature with vigorous shaking. Then, the "detection solution" which consists of coelentrazine (substrate) and Δ11S was premixed, and 80 µL of the detection solution was added to each well. The final concentrations of the substrate, coelentrazine, was 10 µM and the large enzyme fragment, Δ11S, was 500 nM per well in a total volume of 100 µL. Luminescent signal was measured using TECAN Infinite M1000Pro in a kinetic cycle. The results reported here are from the 30-min timepoint.

**Split NanoLuc®-based virus neutralization assay: testing spiked samples and convalescent samples**. Measuring neutralization activity of clinical samples was done in a similar fashion as described above: 10 µL of clinical samples were mixed with 10 pmol of spike protein (5 µL; full S or (S)variant) and 10 pmol of ACE2 fusion (5 µL). All three components were incubated together for 30 min with vigorous shaking. Then, 80 µL of the detection solution (defined above) was added (final total volume per well was 100 µL) and the kinetic cycle of luminescent was measured. For serial dilutions to quantify activity of low titer samples, the assays were performed in a buffer containing 20 mM Tris (pH 7.5), 0.1% Tween-20, 2 mM TCEP, 2 mM EDTA, 25 mM NaCl, and 0.05% BSA. Serum samples (12.5 µL) were first diluted in a volume of 50 µL of buffer, followed by additional one-half serial dilutions up to 6 times. An aliquot (5 µL) of the diluted sample was mixed with 5 µL of β10 modified S binder (20 nM; full S or (S)Variant) and incubated for 30 min. An aliquot (5 µL) of the reaction mixture was further mixed with 45 µL of substrate mixture containing 555.6 nM β9-ACE2-Fc, 100 nM Δ11S, 22.2 µM fur-imazine (substrate). After 1 h incubation, luminescence signals were measured using a microplate reader. The inhibition curve of a sample against S protein variant was obtained by fitting the readings at different dilutions into the normalized response model with variable slope in GraphPad. The titer of 50% neutralization was calculated according to each model.

**Testing spiked samples and convalescent samples with pseudovirus neutralization (PSV) assay**. The active sera (from patients in ICU) and convalescent sera were purified using protein G magnetic beads (Promega Corporation, G7471) as per manufacturer's instruction. The concentrations of purified IgG were measured using NanoDrop 2000. Pseudovirions were produced by co-transfecting 293T human embryonic kidney cells using PEI transfection reagent (Polysciences, Inc., Warrington, PA) with NL4-3 HIV-1 genome (pNL4-3.Luc.R-E-, including the firefly luciferase gene inserted into the nef coding sequence and frameshift mutations in Env and Vpr) and a plasmid encoding the desired virus fusion protein (pCAGGS-SARS2-S-cFlag D614G, kind gift of M. Farzan[47] for SARS-CoV-2 S or pMDG VSV-G for VSV as a specificity control). Forty hours post-transfection, pseudovirus-containing supernatant was filtered (0.45 µM) and concentrated by ultracentrifugation ($26,000 \times g$, 2 h) through a 20% sucrose/TNE (10 mM Tris pH 7.6, 100 mM NaCl, 1 mM EDTA) cushion, and the pellet resuspended in TNE, aliquoted and stored at −80 °C. To measure inhibition of infectivity, 50 µL of $2 \times$ IgG (purified from patient sera) diluted in media was added to CaLu-3 cells (ATCC HTB-55) in a 96-well format, each concentration in triplicate. Fifty microliters of pseudovirus diluted in media + 16 µg/mL DEAE-dextran was added and plates were spinoculated at $2100 \times g$, 30 min, 10 °C. At 20 h, virus and inhibitor were removed via aspiration, and fresh media was replenished. At 40 h, the cells were lysed, and the luciferase activity was measured (Bright-Glo luciferase assay system, Promega, Madison, WI). To determine the normalized luciferase value, average luciferase activity for no virus wells was first subtracted and then the luciferase signals were normalized to the average luciferase activity for no inhibitor wells.

**Protein-based surrogate neutralization assay (pbSNA)**. This assay was performed as previously reported[48]. A 384-well high-binding polystyrene Nunc plates (Thermo Fisher Scientific, #460372) were coated with 100 ng/well of full-length trimeric spike protein provided by the National Research Council Canada (NRC). The plates were centrifuge at $216 \times g$ for 2 min to ensure even coating and incubated overnight with rocking at 4 °C. The serum samples were diluted using a Hamilton MicroLab STAR robotic liquid handler. All plate washing steps included four washes with 100 µL of PBST and were carried using a 405 TS/LS LHC2 plate washer (Biotek Instruments). After the coating step, the plates were washed and blocked for 1 h at room temperature with 80 µL of 3% w/v skim milk powder in PBST. The plates were washed once more and the serum, diluted in 1% w/v skim milk powder in PBST, was added at a final volume of 20 µL per well and incubated with shaking for 2 h. A standard curve of purified neutralizing monoclonal antibody (NRCoV2-20-Fc, NRC) was added alongside a set of pooled negative/positive serum. The plates were then washed and 20 µL of recombinant biotinylated ACE2 (NRC) was added (6.5 ng/well) and incubated with shaking for 1 h. The plates were then washed to remove unbound ACE2 and 20 µL of Streptavidin-Peroxidase Polymer (Sigma #S2438) was added to each well (25 ng/well) and incubated for 1 h

with shaking. Plates were washed one last time and developed by adding 20 µL of ELISA Pico Chemiluminescent Substrate (diluted 1:2 in MilliQ $H_2O$) was dispensed into each well. After a 5-min incubation, plates were read on an Neo2 plate reader (BioTek Instruments) at 20 ms/well and a read height of 1.0 mm. Luminescence signal were blank adjusted, and percentage of ACE2-Spike interaction determined by dividing the luminescence values by the maximal signal (no serum control; maximal ACE2-spike binding; 0% inhibition).

**Statistics and reproducibility**. All data are presented as mean ± standard deviation (SD) and the number of biological repeats, as well as sample size, are reported. No statistical method was performed to predetermine sample size. Statistical analyses were performed using GraphPad Prism 9. Correlation between the Neu-SATiN assay and other assays (PSV or pbSNA) was analyzed using Pearson product–moment correlation.

**Reporting summary**. Further information on research design is available in the Nature Research Reporting Summary linked to this article.

## Data availability

Source data are provided with this paper.

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

## Acknowledgements

We thank the members of Shawn Owen's and Igor Stagljar's lab for their suggestions. This work was supported by the Toronto COVID-19 Action Fund (Connaught Award # 0000313897) to Igor Stagljar and supported by intramural funding from the Office of the Vice President for Research and the 3i Initiative at the University of Utah to Shawn Owen.

## Author contributions

S.J.K. conceptualized the Neu-SATiN assay, was actively involved in experiments and data analysis, and wrote the bulk of the manuscript. Z.Y. was actively involved in designing and preparing probes, conducted experiments and data analysis, and assisted in manuscript editing. M.C.M. was actively involved in many experiments and data analysis and assisted in manuscript writing. D.M.E. and M.S.K. conceptualized and performed the pseudovirus assay and contributed to manuscript editing. A.L. was involved in probe preparation. M.P., A.B., C.B., and P.J. contributed to providing patient samples. Y.G. and M.L. coordinated the protein-based serum neutralization assay. J.C.D. and M.G. coordinated patient samples and performed ELISA neutralization assays. R.A.C. and E.A.M. contributed to providing patient samples. I.S. and S.C.O. guided and supervised the work, contributed to assay development and manuscript writing as well as coordinated the preparation of the manuscript. All authors approved its content.

## Competing interests

A patent covering all the main aspects/key elements of the Neu-SATiN assay has been filed by the governing council of the University of Toronto with the US Patent Office. The application is currently pending. I.S., S.C.O., S.J.K., and Z.Y. are named as coinventor on the patent application. The remaining authors declare no competing interests.
