## [Peer Review File · Nature Communications]

REVIEWER COMMENTS

Reviewer #1 (Remarks to the Author):

This manuscript by Kim et al. reports on the use of a previously published tri-partite complementation assay (SATiN) to measure neutralizing antibodies. The complementation assay was modified compared to the initial publication to now report on the interaction between the Spike protein of SARS-CoV2 and its cognate receptor human ACE2, leading to a decrease in luminescent signal as a function of the concentration of neutralizing antibodies. The resulting assay is straightforward to implement with minimal steps and can be performed directly in patient sera. The benefits of a modular assay are well exemplified here by comparing neutralization potential between different SARS-CoV2 strains. Overall, this is a well-designed study that presents an attractive new method for the rapid evaluation of neutralization potential against SARS-CoV2 variants, a particularly relevant question at this time. It would be great to add new data on the Omicron variant, if available.

Please find below my major comments and questions, followed by some minor remarks.

I.62- “we now show the development of a homogeneous neutralization assay version of SATiN” the authors should quickly summarize how the initial SATiN assay works.

I.109- What is the rationale of tagging the RBD at both termini? How do you explain the discrepancy between mono-tagged and dual-tagged b10 RBD (10x difference in IC50 measurement in Extended Fig 1d) that does not exist with b9?

I.140- “we compared Neu-SATiN to a pseudovirus neutralization (PSV) assay” This is a very important result section but the least compelling one. It is unclear how the Neu-SATiN assay was performed here. In I. 291, it is said that “assays were tested in different matrices” which suggest that the IgG were resuspended in different media. What is the rationale of this? Surely, the Neu-SATiN assay can tolerate cell-culture media and can be strictly compared to the cell neutralization assay...

I.249- “source of discrepancy” give specific examples in the text

Fig.2c- a color-coded figure would be helpful to evaluate the correlation/discrepancy between samples, in relation with the previous point.

Fig.2d/e- For PS6, there is almost no inhibition in fig.2d but there is a 40% inhibition regardless of concentration in Fig.2e. What does that mean? In the case of PS2, inhibition was highly effective in fig. 2d and the PSV assay also identifies it as highly neutralizing but it is not the case in Neu-SATiN in fig 2e. Please comment.

Fig.2e- In the PSV assay, the 10ug/mL data points are often outliers, above the 1ug/mL. Why is that?

Fig.3c- The y-axis scale should be changed to log-scale.

I.287-288- “we show for the first time that monitoring the decreased interaction (...)” This statement is misleading. This is not the first report that showed that decreased Ace2-S binding correlates with neutralization potential (Sancilio AE, et al. Sci Rep. 2021 doi: 10.1038/s41598-021-94653-z for example).

This is not even the first complementation assay to be used for this purpose (see Huang D, et al. J Immunol. 2021 doi: 10.4049/jimmunol.2100155).

The discussion of alternative surrogate assays is lacking here. Elisa assays were mentioned but alternative technologies exist and should be mentioned. In particular, the proximity-based luciferase assay in Huang et al. should be discussed as the technologies are similar. There are also interesting development involving for example FACS (Griffke et al. Journal of Infectious Diseases, 2020, <https://doi.org/10.1093/infdis/jiaa508>), microfluidic (Fiedler S, ACS Infect Dis. 2021 doi: 10.1021/acsinfecdis.1c00047), or biosensors (Huang L, et al. Biosens Bioelectron. 2022 doi: 10.1016/j.bios.2021.113868) that are noteworthy. Overall, the authors should expand and update their discussion of alternative hsVNAs...

I.293- as discussed, SARS-CoV2 variants have different affinities for ACE2, can this be measured by this assay, by titrating ACE2 bit or “cold” ACE2 for example?

I.339- what are the typical yields for the bits? Also for the discussion part, how reproducible is the purification from HEK 293 cells? Can this be standardized enough for repeatability? Is it possible to use other production methods if necessary (e.g. insect cells or cell-free expression)?

I.349- “commercial human serum” what is the provider?

Minor comments/ typos:

I.37- this is misleading as sensitivity was not measured per se. Please rephrase.

I.109- “NABs” to “NAbs”

I.157- “ to stabilize”

I.157- “The rationale for using full-length S protein should provide a more comprehensive measurement of the neutralization effect.”

L241-242- another limitation is that the infective strains are also unknown and could induce different responses

I.360- “ clinical samples were mixed with 10 pmole spike”

Reviewer #2 (Remarks to the Author):

I read with interest the paper “Homogeneous Surrogate 1 Virus Neutralization Assay to Rapidly Assess Neutralization Activity of Anti-SARS-CoV-2 Antibodies” by Kim, Yao, et al. The paper describes a novel method for the detection of biologicals inhibiting the binding of SARS-CoV-2 spike to ACE2 using a

luciferase detector system where key luciferase components b9 and b10 are coupled with ACE2 or spike domains, and measuring how their interaction is inhibited by monoclonal antibodies or polyclonal plasma samples. The authors first show an approach using an RBD binding to the ACE2 to elicit a luciferase signal and benchmark the effects of sera against an HIV-based virus pseudotyped with the SARS-CoV-2 spike molecule. Thereupon, the authors generated a battery of full-length S proteins encoding the luciferase-b10, which covers S from 5 pre-omicron variants of concern. They show the effect of monoclonal antibodies or sera to inhibit this binding and show variant-specific differences in plasma activity against this binding. The paper is informative and the new method has value, but several essential controls need to be provided prior to publication.

They are shown below, in order of decreasing importance:

Major issues

1. The paper does not show a direct comparison of the Neu-SATIN test with a neutralization assay. In figure 2, the authors compared the RBD based Neu-SATIN with a pseudovirus neutralization, but due to restrictions of their neutralization assay, it was not performed on sera or plasma, but rather on the IgG fraction of separated on protein G. The author need to use the same samples in Neu-SATIN and in the PVNT for a direct comparison of efficacies.
2. The paper did not directly compare the assays performed on spike variants in figure 3 and 4 with comparable SARS-CoV-2 variants - or at least pseudoviruses thereof - in a neutralization assay. Hence, it is unclear if the reductions in recognition of VoCs reflect assay-based artifacts or actual reductions in neutralization capacity of antibodies in the sera. Same serum samples need to be tested by Neu-SATIN and a standardly used neutralization assay (e.g. using VSV-based pseudotypes of SARS-CoV-2 spike variants) in order to test them against a benchmark and define if the assay can readily substitute for it. This is especially important because the Neu-SATIN assay works with clinical samples without further processing, and hence a benchmarking neutralization assay that works with such samples is required to ascertain the precision.
3. While it is appreciated that most of the work was done before the advent of Omicron, the paper was submitted after the impact of the new variant was well understood. One of the claims in the paper is that their test is versatile and rapidly adaptable to new variants. Hence, a test with the Omicron Spike variant would demonstrate the validity of this claim. Similar benchmarking as above should be applied.
4. One major claim of the paper is that the Neu-SATIN method is faster and less work-intensive than existing S-Ace2 binding-inhibition assays, and hence an improvement over them. What is the total time needed for Neu-SATIN (not only hands on, but from sample reception to running a 96 well plate), and what exact times are required in assays by the competition? It is not clear if the authors have performed a direct and independent comparison in their lab.

Minor issues:

1. The authors have omitted to discuss some relevant limitations of their approach. The most obvious one is that the assay allows to identify the antibodies that restrict S binding to ACE2, but do not provide information on the antiviral capacity of spike specific antibodies limiting the virus by ATCC or ADP. Furthermore, their approach does not provide information on antibodies from convalescent patients restricting virus replication by targeting other proteins beyond the spike.

2. In Figure 2e the author refer to N=4 but it is unclear if these are technical quadruplicates or something else. They used more than 4 serum samples per assay, so this need a clarification.
3. The same figure legend does not describe what was measured in the pseudovirus neutralization. This information should be indicated in the legend and on the axis, as the readers may expect a plaque assay, but a luciferase reporter assay is described in the method section.
4. The authors have apparently used Spike monomers in Neu-SATIN in figure 3 and 4, rather than the trimeric form that is common in nature, but I am not 100% sure. While this is not a major issue, it needs to be explained better which form was used and which limitations this may impose in detecting the antiviral activity of sera, monoclonals or biologicals, particularly because both a monomeric and a trimeric form are shown in figure 1.
5. In line 155 the authors erroneously state that vaccines are developed to target the full-length S protein in response to the emergence of novel variants. In fact, all currently approved vaccines in US and Canada are based on full-length S protein sequences and have been generated before the onset of VoCs in 2021. This needs to be amended and the reasoning for using the full S ectodomain in Neu-SATIN has to be adapted.
6. Line 367: incubate should be incubated

Reviewer #3 (Remarks to the Author):

Many FDA-authorized serological assays are available to measure prior SARS-CoV-2 infections and COVID19 vaccines, but these assays provide predominantly yes/no answers or, at best, semi-quantitative results on the levels of Spike- or Nucleocapsid-specific antibodies. These assays do not measure function, nor do they account for time-post infection/vaccination or affinity maturation. As simple seroprevalence studies become less and less relevant, new assays are needed to measure antibody function as a true surrogate of immunity. In turn, these assays can help estimate susceptibility in populations and the need for boosters at an individual level.

In this manuscript by Kim...Owen and colleagues, a new method is described to measure SARS-CoV-2-neutralizing antibodies. In this approach, a split luciferase system is used to tag peptides to ACE2 and Spike, which can then be complemented by the remaining piece of luciferase. Enzymatic activity is only observed upon binding of ACE2 to Spike. In the presence of neutralizing antibodies, this interaction is prevented and luciferase activity is suppressed. The authors optimize a variety of combinations of tags to the N- and C-termini of Spike and ACE2 and demonstrate that the assay can be readily adapted to

variants of concern. The results correlate well with pseudovirus assays and perform as expected with well-characterized monoclonal antibodies. Given that live or pseudovirus neutralizations are rarely used in clinical tests, to my knowledge the only real alternative is the sVNT assay reported by Tan and colleagues (Nature Biotech, 2020). The authors report that the workflow here is substantially faster and simpler than the sVNT assay.

Overall, this is a clearly written manuscript that reports highly relevant results and methods that will be useful to many groups, both for clinical diagnostics and for research. I do have a few technical concerns.

1) I am unclear why IgG was purified for the pseudovirus neutralization assays in Figure 2. This is not typically done for these assays, and as the authors mention, the contribution of IgM and IgA are excluded if protein G is used to purify. These assays should be repeated using serum or plasma that is serially diluted.

2) Using pseudovirus neutralization assays as a standard is not ideal. There is a wide degree of variability between groups as to how well these assays correlate with live virus neutralization (e.g. Abayasingam et al., Cell Rep Med, 2021; Ripperger et al, Immunity, 2020; Wang...Ho, Nature, 2020). If a BSL3 facility is available, Neu-Satin should be compared to live virus neutralization.

3) Related to the above point, there are several highly potent neutralizing antibodies that do not interfere with Spike-ACE2 attachment. For example, sotrovimab, the only FDA-authorized mAb that remains fully potent against Omicron, does not prevent RBD-ACE2 binding (Pinto et al., Nature 2020). Comparisons with robust live virus neutralization assays would provide a sense of how much Neu-Satin underestimates neutralizing antibodies in sera/plasma. This will become increasingly important if antibodies against conserved neutralizing epitopes, which will be preferentially recalled in subsequent exposures, do not block Spike-ACE2 interactions.

REVIEWER COMMENTS

Reviewer #1 (Remarks to the Author):

This manuscript by Kim et al. reports on the use of a previously published tri-partite complementation assay (SATiN) to measure neutralizing antibodies. The complementation assay was modified compared to the initial publication to now report on the interaction between the Spike protein of SARS-CoV2 and its cognate receptor human ACE2, leading to a decrease in luminescent signal as a function of the concentration of neutralizing antibodies. The resulting assay is straightforward to implement with minimal steps and can be performed directly in patient sera. The benefits of a modular assay are well exemplified here by comparing neutralization potential between different SARS-CoV2 strains. Overall, this is a well-designed study that presents an attractive new method for the rapid evaluation of neutralization potential against SARS-CoV2 variants, a particularly relevant question at this time. It would be great to add new data on the Omicron variant, if available.

We appreciate the reviewer's evaluation of our report and for the detailed questions and suggestions for improvement. We provide a point-by-point response to these concerns below, but first highlight that we have added data on the Omicron Variant (included in Figure 3 e and f, as well as, Figure 4 b and c).

Please find below my major comments and questions, followed by some minor remarks.

I.62- "we now show the development of a homogeneous neutralization assay version of SATiN" the authors should quickly summarize how the initial SATiN assay works.

The following sentences were added to address reviewer's comment (see : The initial SATiN utilized spike protein and protein G tagged with either b9 or b10. Upon simultaneous binding of the tagged spike protein and protein G to anti-SARS-CoV-2 antibody, b9 and b10 were brought into proximity and allowed refolding of Δ 11S into active luciferase producing glow-type luminescence.

I.109- What is the rationale of tagging the RBD at both termini? How do you explain the discrepancy between mono-tagged and dual-tagged b10 RBD (10x difference in IC50 measurement in Extended Fig 1d) that does not exist with b9?

This is an insightful question. When designing the binders, we hypothesized that putting more than one tag would increase the chance of proximity between b10 and b9, leading to more Δ 11S refolding, and therefore producing more luminescence; however, we have recently found that distance between binders/tags is not the only factor that influences complementation, but likely the orientation of the peptide tags has an impact on the ease of complementation (see SJ Kim, AS Dixon, SC Owen Acta Biomaterialia 135, 225-233). As such, we posit that b10 at both termini provide a better alignment than b9. Although there is not always an improvement in IC50/signal with a dual-tagged approach, it is important to note that it is not detrimental.

I.140- "we compared Neu-SATiN to a pseudovirus neutralization (PSV) assay" This is a very important result section but the least compelling one. It is unclear how the Neu-SATiN assay was performed here. In I. 291, it is said that "assays were tested in different matrices" which suggest that the IgG were resuspended in different media. What is the rationale of this? Surely, the Neu-SATiN assay can tolerate cell-culture media and can be strictly compared to the cell neutralization assay

Thank you for bringing these issues to our attention. Due to the sensitivity of this initial PSV assay, purified IgG must be diluted in cell culture compatible media. In contrast, the Neu-SATiN assay can be run directly from serum/plasma samples which are diluted into simple buffers, circumventing the need for IgG purification. Our goal in this study was to test the feasibility of the Neu-SATiN assay in correlation with a more common assay type. We have modified Figure 2 to make the results clearer and more concise and moved the PSV assay results to Extended Data Figure 1 to allow more explanation .

I.249- “source of discrepancy” give specific examples in the text

We are not sure what exactly the reviewer is highlighting as I.249 does not contain this phrase, but we assume the reviewer is asking about this statement on I.154. We included the lack of IgMs in the purified samples as one possible source of discrepancy between the assays. We apologize if we unintentionally misunderstood the concern.

Fig.2c- a color-coded figure would be helpful to evaluate the correlation/discrepancy between samples, in relation with the previous point.

As Figure 2c does not compare samples, we assume that the reviewer instead means Figure 2f. We agree that changing the presentation of this data could be improved for clarity. As indicated above, we have changed Figure 2 d-f to provide a clearer summary of the results, including color-coding the figure as suggested. We also provide a side-by-side comparison of sample results between the two assays on the least diluted values. We highlight these values because values for further dilutions in both assays are low. We have still provided the full dilution data. Importantly, we have expanded our comparison of the full spike (trimeric) version of Neu-SATiN to an established surrogate neutralization assay to further demonstrate the correlation between platforms (see Figure 3 for Delta and Extended Figure 4 for Omicron comparisons, respectively).

Fig.2d/e- For PS6, there is almost no inhibition in fig.2d but there is a 40% inhibition regardless of concentration in Fig.2e. What does that mean? In the case of PS2, inhibition was highly effective in fig. 2d and the PSV assay also identifies it as highly neutralizing but it is not the case in Neu-SATiN in fig 2e. Please comment.

The difference is most likely due to presence/absence of IgM. For example, we think that PS 6 does have neutralizing IgGs but perhaps the titer is low and have almost no IgMs, therefore the effect was not well observed in Fig 2d when straight plasma was used. However, after purifying IgGs out from the PS 6 plasma and concentrating them, we were able to see 40% inhibition in Fig 2e. In case of PS 2, plasma of PS 2 perhaps has high titer of both IgGs and IgMs, therefore showed high inhibition in Fig 2d. However, in Fig 2e, as only IgGs were used, the inhibition is not as effective compared to Fig 2d.

Fig.2e- In the PSV assay, the 10ug/mL data points are often outliers, above the 1ug/mL. Why is that?

The effect of diluted antibodies used in the PSV are likely below limit of detection and the normalization process commonly produces this kind of trend. Considering the low response of samples at these dilutions, we have opted to instead provide a summary of the results at the most concentrated samples (as described above).

Fig.3c- The y-axis scale should be changed to log-scale.

Unfortunately, a log-scale cannot display 0 µg/mL.

I.287-288- “we show for the first time that monitoring the decreased interaction (...)” This statement is misleading. This is not the first report that showed that decreased Ace2-S binding correlates with neutralization potential (Sancilio AE, et al. Sci Rep. 2021 doi: 10.1038/s41598-021-94653-z for example). This is not even the first complementation assay to be used for this purpose (see Huang D, et al. J Immunol. 2021 doi: 10.4049/jimmunol.2100155). The discussion of alternative surrogate assays is lacking here. Elisa assays were mentioned but alternative technologies exist and should be mentioned. In particular, the proximity-based luciferase assay in Huang et al. should be discussed as the technologies are similar. There are also interesting development involving for example FACS (Griffke et al. Journal of Infectious Diseases, 2020, <https://doi.org/10.1093/infdis/jiaa508>), microfluidic (Fiedler S, ACS Infect Dis. 2021 doi:

10.1021/acsinfecdis.1c00047), or biosensors (Huang L, et al. Biosens Bioelectron. 2022 doi: 10.1016/j.bios.2021.113868) that are noteworthy. Overall, the authors should expand and update their discussion of alternative hsVNAs...

Although we did attempt to both highlight and distinguish our platform from other reports (e.g. ref 10), we appreciate the reviewer's concern in making sure that we are transparent. We have added/amended the following paragraph in the discussion section:

There have been several reports describing efforts to develop surrogate virus neutralization (sVNA) assays for the detection of antibodies against SARS-CoV-2³⁷⁻⁴⁰. These reports show capabilities as surrogate assays; however, many of these platforms require chemical modification and conjugation of proteins to beads or other solid surfaces, requiring intermediate purification and/or washing steps to remove unbound reagents and analytes. In contrast, Huang et al. reported the development of an hsVNA using the binary version of split-NanoLuc[®]. Although the approach is similar to the platform we report here, the authors used a monomeric or dimeric form of S1 domain of spike protein tagged with b10 (SmBiT), paired with ACE2 (LgBiT, NanoLuc[®] beta strands 1 through 9)³⁷, which was shown to have limited detection of non-RBD targeting antibodies. The importance of conformational changes in spike protein altering ACE2 binding has been emphasized in many studies^{21, 41}. Also, with new variants of concern emerging, many of the deployed vaccines and second-wave vaccines in development are targeting full spike protein⁴²⁻⁴⁴. We initially designed our assay by investigating the interactions between ACE2 and RBD¹⁰. The result is in agreement with reports that neutralization activities of NAb mostly come from anti-RBD immunoglobulins^{35, 45}; however, immunologic response can also produce antibodies against epitopes outside of the RBD and may require the trimeric structure of the full spike protein for binding. This compelled us to develop stabilized, full-length-spike (both WT and major variants of concern) versions of the binders and stable, dimerized ACE2 receptor. The data presented in Fig. 3 demonstrates that b10-(S)WT/ b9-ACE2-Fc can differentiate the degree of neutralization directly from patient sera and that variant versions of the spike binders can distinguish anti-SARS-CoV-2 NAb positive serum samples from negative samples (Fig 3e,f). In addition, Neu-SATiN can provide quantitative analysis of NT50 and thus enables the measurement of potency of anti-SARS-CoV-2 antibodies against different strains (Fig 4). Combined, this provides a comprehensive screen of a patient's level of protection against the current variants of concern. As the assay is modular, emerging variants of interest can be quickly produced and incorporated as we have demonstrated with the inclusion of the Omicron variant that become prominent during the initial review of this manuscript.

I.293- as discussed, SARS-CoV2 variants have different affinities for ACE2, can this be measured by this assay, by titrating ACE2 bit or "cold" ACE2 for example?

We appreciate the insightful suggestion. In our previous publication (Dixon et al. Scientific Reports. 2017), using HER2 binders, we have done a kinetic study similar to what the review is describing here. Measuring the affinities between SARS-CoV-2 variants and ACE2 is feasible and we provide the resulting data here for the reviewer's benefit. We hesitate to include this data as part of the manuscript because there are no reports that directly correlate binding affinity and infectivity and we do not wish to mislead readers. We are open to insight from the reviewer on this matter.

	WT	Alpha	Beta	Gamma	Delta	Omicron
KD (nM)	27.77	16.04	29.91	53.37	205	90.31
SD	1.08	0.9973	1.05	1.156	2.012	1.446

I.339- what are the typical yields for the bits? Also for the discussion part, how reproducible is the purification from HEK 293 cells? Can this be standardized enough for repeatability? Is it possible to use other production methods if necessary (e.g. insect cells or cell-free expression)?

We have been able to successfully reproduced the binders multiple times in μM range (typically between 5-10 μM) using these CRISPR engineered HEK293 cells. Proper posttranslational modifications are needed for correct S protein folding and therefore E. coli and cell-free systems are not suitable for this purpose. The usability of insect expression system in this production needs to be verified. We have not tried insect cells or cell-free expression systems.

I.349- “commercial human serum” what is the provider? Vendor and catalog information added.

Minor comments/ typos:

I.37- this is misleading as sensitivity was not measured per se. Please rephrase. We changed the word “sensitivity” to “correlation”

I.109- “NABs” to “NAbs” – typo fixed

I.157- “ to stabilize” – typo fixed

I.157- “The rationale for using full-length S protein should provide a more comprehensive measurement of the neutralization effect.” Sentence rewritten

L241-242- another limitation is that the infective strains are also unknown and could induce different responses. This comment was added to the manuscript

I.360- “ clinical samples were mixed with 10 pmole spike” – type fixed

Reviewer #2 (Remarks to the Author):

I read with interest the paper “Homogeneous Surrogate 1 Virus Neutralization Assay to Rapidly Assess Neutralization Activity of Anti-SARS-CoV-2 Antibodies” by Kim, Yao, et al. The paper describes a novel method for the detection of biologicals inhibiting the binding of SARS-CoV-2 spike to ACE2 using a luciferase detector system where key luciferase components b9 and b10 are coupled with ACE2 or spike domains, and measuring how their interaction is inhibited by monoclonal antibodies or polyclonal plasma samples. The authors first show an approach using an RBD binding to the

ACE2 to elicit a luciferase signal and benchmark the effects of sera against an HIV-based virus pseudotyped with the SARS-CoV-2 spike molecule. Thereupon, the authors generated a battery of full-length S proteins encoding the luciferase-b10, which covers S from 5 pre-omicron variants of concern. They show the effect of monoclonal antibodies or sera to inhibit this binding and show variant-specific differences in plasma activity against this binding. The paper is informative and the new method has value, but several essential controls need to be provided prior to publication.

We thank the reviewer for their evaluation of our manuscript and we provide a point-by-point response to concerns below.

They are shown below, in order of decreasing importance:

Major issues

1. The paper does not show a direct comparison of the Neu-SATIN test with a neutralization assay. In figure 2, the authors compared the RBD based Neu-SATIN with a pseudovirus neutralization, but due to restrictions of their neutralization assay, it was not performed on sera or plasma, but rather on the IgG fraction of separated on protein G. The author need to use the same samples in Neu-SATIN and in the PVNT for a direct comparison of efficacies.

We understand the reviewer's concern and now provide a direct comparison of Neu-SATiN with an established benchmarking surrogate neutralization assay (see reference 49). Working with collaborators at the University of Ottawa (Yannick Galipeau and Marc-André Langlois), the neutralization from a majority of samples (N=66) shown in both Figures 3 and 4 were assessed and the level of activity compared to SATiN. The full results of the surrogate neutralization assay are provided in Extended Figure 3 and comparisons to Neu-SATiN provided in Figure 3 for Delta and Extended Figure 4 for Omicron.

Reviewer 1 has a similar concern regarding the RBD version of Neu-SATiN compared to a PSA. As discussed, the sensitivity of the PSV assay (results in Extended Figure 1) required that purified IgG must be used. Our goal in that pilot study was to test the feasibility of the Neu-SATiN assay in correlation with a more common assay type and we have modified Figure 2 to make the results clearer and more concise. Since we use the full-spike constructs for all other analyses, and these are performed on a larger (and more characterized) set of clinical samples, we have opted not to repeat that study.

2. The paper did not directly compare the assays performed on spike variants in figure 3 and 4 with comparable SARS-CoV-2 variants - or at least pseudoviruses thereof - in a neutralization assay. Hence, it is unclear if the reductions in recognition of VoCs reflect assay-based artifacts or actual reductions in neutralization capacity of antibodies in the sera. Same serum samples need to be tested by Neu-SATIN and a standardly used neutralization assay (e.g. using VSV-based pseudotypes of SARS-CoV-2 spike variants) in order to test them against a benchmark and define if the assay can readily substitute for it. This is especially important because the Neu-SATIN assay works with clinical samples without further processing, and hence a benchmarking neutralization assay that works with such samples is required to ascertain the precision

We appreciate the reviewer's request. As mentioned above, we now include a comparison of neu-SATiN against a standard of the wild-type (Wuhan) variant. It is most prudent to compare to the WT variant because of the clinical characterization of our samples. The limited sample volume of our clinical samples prevents analysis for each of the variants; however, we have also performed a comparative study with the Omicron variant.

3. While it is appreciated that most of the work was done before the advent of Omicron, the paper was submitted after the impact of the new variant was well understood. One of the claims in the paper is that their test is versatile and rapidly adaptable to new variants. Hence, a test with the Omicron Spike variant would demonstrate the validity of this claim. Similar benchmarking as above should be applied.

We constructed the Omicron Spike variant and have updated all graphs with the relevant data. The following statement was amended to highlight the inclusion of the new variant:

As the assay is modular, emerging variants of interest can be quickly produced and incorporated as we have demonstrated with the inclusion of the Omicron variant that became prominent during the initial review of this manuscript.

4. One major claim of the paper is that the Neu-SATIN method is faster and less work-intensive than existing S-Ace2 binding-inhibition assays, and hence an improvement over them. What is the total time needed for Neu-SATIN (not only hands on, but from sample reception to running a 96 well plate), and what exact times are required in assays by the competition? It is not clear if the authors have performed a direct and independent comparison in their lab.

The most time-consuming parts of Neu-SATIN is to make CRISPR engineered HEK293 cells and expressing and purifying the binders. However, these steps can be more facilitated by biotech companies. Once all of the reagents are ready, actual time from sample mixing to plate reading (a 96 well plate) only takes about 30 min to an hour at max. Although we have not personally tested other assays to compare, there are many review papers that highlight different assay times (e.g., References 1 and 3). We've used these references to claim short assay time as one of the strengths of Neu-SATIN.

Minor issues:

1. The authors have omitted to discuss some relevant limitations of their approach. The most obvious one is that the assay allows to identify the antibodies that restrict S binding to ACE2, but do not provide information on the antiviral capacity of spike specific antibodies limiting the virus by ATCC or ADP. Furthermore, their approach does not provide information on antibodies from convalescent patients restricting virus replication by targeting other proteins beyond the spike.

We appreciate the importance of clarifying that the assay limitations and have added the following as part of the discussion:

It is important to note that natural immunity can produce antibodies that bind several antigens and function through alternate mechanisms, including antibody-dependent phagocytosis (ADP) or antibody-dependent cellular cytotoxicity (ADCC). Although the current format of Neu-SATIN cannot measure these types of antiviral activity, the majority of vaccines use Spike domains^{17,18}; therefore, this assay can be used to assess protection developed from immunization.

2. In Figure 2e the author refer to N=4 but it is unclear if these are technical quadruplicates or something else. They used more than 4 serum samples per assay, so this need a clarification.

We understand the confusion. N=4 indicates that each assay (Neu-SATIN or PSV) was repeated four individual times. Every time when the assay was done, the samples were run in technical triplicates and averaged. The figure legend was edited to make it clearer.

3. The same figure legend does not describe what was measured in the pseudovirus neutralization. This information should be indicated in the legend and on the axis, as the readers may expect a plaque assay, but a luciferase reporter assay is described in the method section.

We have updated the figure legend with more detailed information.

4. The authors have apparently used Spike monomers in Neu-SATIN in figure 3 and 4, rather than the trimeric form that is common in nature, but I am not 100% sure. While this is not a major issue, it needs to be explained better which form was used and which limitations this may impose in detecting the antiviral activity of sera, monoclonals or biologicals, particularly because both a monomeric and a trimeric form are shown in figure 1.

We appreciate the reviewer for highlighting this confusion. We understand that the trimeric form is more stable than the monomeric form; however, the trimeric form of spike protein needs to undergo conformational changes mediated by TM protease serin 2 (TMPRSS2) in order to put RBD in “open” position to then interact with ACE2. We recognize that this protease is one of the potential drug targets for SARS-CoV-2 therapy and our assay may be limited only to screening antibodies. Nevertheless, we agree that there are important structural components that omitted by using a fragment of the Spike (e.g. RBD or S1 only) and have further explained our motivation to use the entire spike protein in a trimeric form. We also highlight this distinction of our assay in our discussion section (please see response to Reviewer #1 requesting comparison to other assays).

5. In line 155 the authors erroneously state that vaccines are developed to target the full-length S protein in response to the emergence of novel variants. In fact, all currently approved vaccines in US and Canada are based on full-length S protein sequences and have been generated before the onset of VoCs in 2021. This needs to be amended and the reasoning for using the full S ectodomain in Neu-SATIN has to be adapted.

We value the suggestion be the reviewer and have added the following sentence to the manuscript:

In fact, all currently approved vaccines in US and Canada are based on full-length S protein sequences and have been generated before the onset of variants in 2021.

6. Line 367: incubate should be incubated **typo fixed**

Reviewer #3 (Remarks to the Author):

Many FDA-authorized serological assays are available to measure prior SARS-CoV-2 infections and COVID19 vaccines, but these assays provide predominantly yes/no answers or, at best, semi-quantitative results on the levels of Spike- or Nucleocapsid-specific antibodies. These assays do not measure function, nor do they account for time-post infection/vaccination or affinity maturation. As simple seroprevalence studies become less and less relevant, new assays are needed to measure antibody function as a true surrogate of immunity. In turn, these assays can help estimate susceptibility in populations and the need for boosters at an individual level.

In this manuscript by Kim...Owen and colleagues, a new method is described to measure SARS-CoV-2-neutralizing antibodies. In this approach, a split luciferase system is used to tag peptides to ACE2 and Spike, which can then be complemented by the remaining piece of luciferase. Enzymatic activity is only observed upon binding of ACE2 to Spike. In the presence of neutralizing antibodies, this interaction is prevented and luciferase activity is suppressed. The authors optimize a variety of combinations of tags to the N- and C-termini of Spike and ACE2 and demonstrate that the assay can be readily adapted to variants of concern. The results correlate well with pseudovirus assays and perform as expected with well-characterized monoclonal antibodies. Given that live or pseudovirus neutralizations are rarely used in clinical tests, to my knowledge the only real alternative is the sVNT assay reported by Tan and colleagues (Nature Biotech, 2020). The authors report that the workflow here is substantially faster and simpler than the sVNT assay.

Overall, this is a clearly written manuscript that reports highly relevant results and methods that will be useful to many groups, both for clinical diagnostics and for research. I do have a few technical concerns.

We thank the reviewer for their evaluation of our manuscript and we provide a point-by-point response to concerns below.

1) I am unclear why IgG was purified for the pseudovirus neutralization assays in Figure 2. This is not typically done for

these assays, and as the authors mention, the contribution of IgM and IgA are excluded if protein G is used to purify. These assays should be repeated using serum or plasma that is serially diluted.

Reviewers 1 and 2 have a similar concern regarding the PSV assay (results in Fig 2). The sensitivity of this HIV-based PSA requires that purified IgG must be used. Our goal in that pilot study was to test the feasibility of the Neu-SATiN assay in correlation with a common assay type and we have modified Figure 2 to make the results clearer and more concise. We now include an additional surrogate neutralization assay to analyze clinical samples used in Fig 3 and Fig 4 studies. Serum was used for the newly reported studies.

2) Using pseudovirus neutralization assays as a standard is not ideal. There is a wide degree of variability between groups as to how well these assays correlate with live virus neutralization (e.g. Abayasingam et al., Cell Rep Med, 2021; Ripperger et al, Immunity, 2020; Wang...Ho, Nature, 2020). If a BSL3 facility is available, Neu-Satin should be compared to live virus neutralization.

Although pseudovirus assays and surrogate neutralization assays cannot fully recapitulate a live virus, these are well-accepted surrogates. Access to live virus is extremely limited and prohibitive.

3) Related to the above point, there are several highly potent neutralizing antibodies that do not interfere with Spike-ACE2 attachment. For example, sotrovimab, the only FDA-authorized mAb that remains fully potent against Omicron, does not prevent RBD-ACE2 binding (Pinto et al., Nature 2020). Comparisons with robust live virus neutralization assays would provide a sense of how much Neu-Satin underestimates neutralizing antibodies in sera/plasma. This will become increasingly important if antibodies against conserved neutralizing epitopes, which will be preferentially recalled in subsequent exposures, do not block Spike-ACE2 interactions.

We agree that there are antiviral mechanisms that do not rely only on blocking the RBD – ACE2 interaction but through binding a non-RBD epitope on the spike protein. In fact, this was one of our main motivations in switching to a full-spike construct. We also appreciate that there are antibodies that prevent or fight infection through alternate mechanisms. To address these possibilities in the context of limitations of the assay, we have added the following to the discussion section:

It is important to note that natural immunity can produce antibodies that bind several antigens and function through alternate mechanisms, including antibody-dependent phagocytosis (ADP) or antibody-dependent cellular cytotoxicity (ADCC). Although the current format of Neu-SATiN cannot measure these types of antiviral activity, the majority of vaccines use Spike domains^{16,17}; therefore, this assay can be used to assess protection developed from immunization.

REVIEWERS' COMMENTS

Reviewer #1 (Remarks to the Author):

The revised manuscript answers most of the questions raised by the initial one. It presents a very valuable tool that can quickly adapt to the emergence of new variants without losing efficiency. The manuscript is well written and the conclusions are well supported.

Minor comments:

l.174+ fig 3b- why was Omicron not included here, even though the construct was obtained?

l.199-a succinct description of the protein-based neutralization assay would be helpful

fig3 f/g- the tested samples present very little neutralization of Omicron. It would be good to provide some information on these samples as it may help with data interpretation. Indeed, if the samples were collected before the omicron dominance, this shows that previous infection by delta does not protect against omicron. If the samples were collected later, then omicron does not elicit much immune response, even against itself...

Reviewer #2 (Remarks to the Author):

The authors have addressed all of my comments. Many thanks for the reply.

Reviewer #3 (Remarks to the Author):

The authors explained the need to purify IgG for the pseudovirus neutralization assays. A specific explanation was not given as to why live virus neutralization assays could not be done--a web search does reveal a BSL3 facility and virology core at Utah. Nonetheless, this does represent an alternative to the sVNT assays reported by Tan et al., Nat Biotech 2020, that could be useful for clinical and functional antibody testing.

We appreciate the editors' and reviewers' evaluation and support of our revised manuscript. Below, we provide a point-by-point response to the remaining minor concerns.

Reviewer #1 (Remarks to the Author):

The revised manuscript answers most of the questions raised by the initial one. It presents a very valuable tool that can quickly adapt to the emergence of new variants without losing efficiency. The manuscript is well written and the conclusions are well supported.

We thank the reviewer for their summary and insight.

Minor comments:

I.174+ fig 3b- why was Omicron not included here, even though the construct was obtained?

The omicron data was unintentionally omitted and Figure 3b has been updated to include the information.

I.199-a succinct description of the protein-based neutralization assay would be helpful

We have added the following to the manuscript: The protein-based neutralization assay is performed by coating microwells with recombinant SARS-CoV-2 Spike trimer and then adding plasma from patient samples. An ACE2-HRP construct is used to assess the presence of neutralizing antibodies that prevent binding of ACE2 to Spike trimer.

fig3 f/g- the tested samples present very little neutralization of Omicron. It would be good to provide some information on these samples as it may help with data interpretation. Indeed, if the samples were collected before the omicron dominance, this shows that previous infection by delta does not protect against omicron. If the samples were collected later, then omicron does not elicit much immune response, even against itself...

We thank the reviewer for this excellent suggestion to clarify some information on samples and related interpretation of the results. As the reviewer notes, the samples were collected well-before Omicron and we agree that our results suggest that previous vaccination/infection provides little neutralizing efficacy against Omicron. We have added the statements into the manuscript and Figure 3 caption:

(~Line 288) As all of the patient samples were collected before November 2021, prior to any known wide-spread Omicron infections, our results also suggest that prior infection/vaccination does not provide significant neutralizing protection against the variant.

(Figure 3 caption)). All patient samples were collected before 2021.11, prior to any known Omicron infections.

Reviewer #2 (Remarks to the Author):

The authors have addressed all of my comments. Many thanks for the reply.

We are glad that the reviewer's comments were addressed.

Reviewer #3 (Remarks to the Author):

The authors explained the need to purify IgG for the pseudovirus neutralization assays. A specific explanation was not given as to why live virus neutralization assays could not be done--a web search does reveal a BSL3 facility and virology core at Utah. Nonetheless, this does represent an alternative to the sVNT assays reported by Tan et al., Nat Biotech 2020, that could be useful for clinical and functional antibody testing.

We understand the reviewer's comments and point out that the single BSL3 facility at the University of Utah is currently in transition and has only ever been certified for HIV work.